# Comprehensive Analysis of HIV-1 Integrase Resistance-Related Mutations in African Countries

**DOI:** 10.3390/pathogens13020102

**Published:** 2024-01-24

**Authors:** Francesco Branda, Marta Giovanetti, Leonardo Sernicola, Stefania Farcomeni, Massimo Ciccozzi, Alessandra Borsetti

**Affiliations:** 1Unit of Medical Statistics and Molecular Epidemiology, University Campus Bio-Medico of Rome, 00128 Rome, Italy; f.branda@unicampus.it (F.B.); m.ciccozzi@unicampus.it (M.C.); 2Sciences and Technologies for Sustainable Development and One Health, Università Campus Bio-Medico di Roma, 00128 Rome, Italy; giovanetti.marta@gmail.com; 3Instituto Rene Rachou, Fundação Oswaldo Cruz, Belo Horizonte 30190-002, Brazil; 4Climate Amplified Diseases and Epidemics (CLIMADE), Brasilia 70070-130, Brazil; 5National HIV/AIDS Research Center (CNAIDS), Istituto Superiore di Sanità, 00161 Rome, Italy; leonardo.sernicola@iss.it (L.S.); stefania.farcomeni@iss.it (S.F.)

**Keywords:** HIV-1, integrase, drug resistance, viral evolution, African countries

## Abstract

The growing emergence of non-nucleoside reverse transcriptase inhibitor (NNRTI) HIV drug resistance in sub-Saharan Africa (SSA) led to the World Health Organization (WHO) recommending, in 2018, a transition to dolutegravir (DTG) as a first-line antiretroviral therapy (ART) in SSA. The broad HIV-1 genetic diversity in SSA could shape DTG effectiveness and the pattern of drug resistance mutations (DRMs) in this region. This study evaluated HIV-1 integrase (IN) DRMs and conserved regions among published groups M, N, O, and P HIV-1 sequences spanning forty years of the HIV epidemic during the transition of DTG-based ART. Overall, we found low levels of integrase strand transfer inhibitor (INSTI)-DRMs (<1%) across HIV groups between the years 1983 and 2023; however, it was unexpected to detect DRMs at statistically significantly higher frequencies in pre-INSTI (1983–2007) than in the INSTI (2008–2023) era. The variability of accessory INSTI-DRMs depended on the HIV subtypes, with implications for susceptibility to DTG. Our findings provide new perspectives on the molecular epidemiology and drug resistance profiles of INSTIs in SSA, emphasizing the need for ongoing surveillance and customized treatment approaches to address the continent’s varied HIV subtypes and changing resistance patterns.

## 1. Introduction

At the close of 2023, approximately 39 million individuals globally were living with HIV (PLWH) [WHO], and Africa, with 25.6 million infected people, accounting for the highest HIV-1 genetic diversity globally, exhibits different distributions of HIV-1 subtypes and recombinant forms [1,2]. This includes subtype B in Northern Africa, subtype C in the South, CFR02_AG in the West and Central, and subtype C and D in the East [1,2,3,4]. The extensive genetic diversity of HIV, especially in West and Central Africa, potentially contributes to the emergence of natural polymorphisms associated with resistance and varying susceptibilities to combined antiretroviral therapy (cART) [5]. While cART has significantly reduced transmission, morbidity, and mortality among People Living with HIV (PLWH), the rise of drug resistance mutations (DRMs) in the virus increases the risk of virological failure and challenges the effectiveness of cART [5].

In many countries of Sub-Saharan Africa (SSA), high levels of pretreatment drug resistance to nucleotide reverse transcriptase inhibitors (NRTIs), non-nucleoside reverse transcriptase inhibitors (NNRTIs), and protease inhibitors (PIs) have led, in 2018, the World Health Organization (WHO) to recommend a transition to dolutegravir (DTG)—an integrase strand transfer inhibitor (INSTI) with a strong resistance barrier—as a first-line ART component in resource-limited settings [6]. INSTIs, targeting the integrase (IN) enzyme crucial for HIV replication, have shown high efficacy in treatment-naïve and -experienced individuals [7]. The first INSTI approved by the United States Food and Drug Administration (US FDA) for clinical use was raltegravir (RAL) in 2007, followed by elvitegravir (EVG) in 2012, dolutegravir (DTG) in 2013, bictegravir (BIC) in 2018, and cabotegravir (CAB) in 2021 [6,7,8].

In African countries with prevalent NRTI resistance, a transition to (DTG)-based ART is underway. Optimized use of DTG is vital for cART success in settings lacking genotyping testing or drug resistance surveillance to prevent the emergence of resistant variants. The Stanford HIV Drug Resistance Database (HIVdb) (http://hivdb.stanford.edu, accessed on 25 November 2023). identifies key IN mutations across HIV-1 subtypes at positions T66, E92, G118, E138, G140, Y143, S147, Q148, N155 and R263 [9,10,11,12]. Additionally, mutations that reduce susceptibility in combination with major DRMs include polymorphic M50, L74, T97 and nonpolymorphic H51, Q95, F121, Q146, G149, V151, S153, G163, S230, and D232 accessory mutations. The pattern of DRMs varies among HIV-1 subtypes, like the G118R mutation in non-B subtypes and R263K in subtype B [9,10,11,12]. Given the significant INSTI susceptibility differences among HIV subtypes and the genetic diversity in SSA, it is crucial to monitor and investigate the pattern of INSTI-DRMs through a surveillance program. This study aims to provide updated insights into the molecular epidemiology and drug resistance mutation profiles of INSTIs in published IN sequences from 1983 to 2023 of the four phylogenetic M, N, O, and P HIV-1 groups in African countries.

## 2. Materials and Methods

### 2.1. Spatio-Temporal HIV-1 Subtype Epidemic Trend

We retrieved 7906 HIV-1 IN sequences, covering all 864 base pairs, collected from both ART-naïve and ART-treated individuals between 1983 and 2022 from a public database maintained by Los Alamos National Laboratory https://www.hiv.lanl.gov/content/index (accessed on 1 October 2023). HIV-1 group M included sequences belonging to subtypes A, B, C, D, F, G, H, J, K, L, and the circulating recombinant forms (CRFs) 01 to 143. A set of 96 sequences referring to HIV-1 groups N, O, P, and untyped “U” were assembled separately from the M group. The temporal distribution of HIV-1 IN subtypes across the SSA was meticulously analyzed through a comprehensive study of these 7906 HIV-1 samples. The selection criteria for samples were stringent, prioritizing those with detailed epidemiological metadata, including the date of sample collection and the country of origin. These HIV-1 strains represent a diverse set of geographical origins, categorized into four major African macro-regions: Western, Eastern, Central, and Southern. This regional classification followed the framework provided by UN-AIDS. The study further explored the nuances of subtype distribution within these regions, revealing patterns and trends indicative of both the biological characteristics of the virus and the socio-epidemiological factors at play.

### 2.2. Analysis of Polymorphisms and Integrase DRMs

All sequences were submitted to the Stanford HIV Drug Resistance Database version 7.0 for quality analysis. Sequence quality control was ensured by excluding any sequence having two or more of the following features: internal stop codons, frame shifts, more than 1% ambiguous nucleotide, hypermutations, insertions, or deletions. To assess the DRM_s_, we used the Calibrated Population Resistance (CPR) tool version 8.1, available at Stanford University HIV Drug Resistance Database (HIVdb) (https://hivdb.stanford.edu/, (accessed on 1 October 2023)) and Stanford HIVDB 8.9-1 “https://hivdb.stanford.edu/dr-summary/comments/INSTI/ (last updated 25 November 2023)”. The analysis also employed the 2022 IAS–USA DRM list using the HIV-1 subtype B reference sequence (GenBank accession number: K03455).

### 2.3. Examination of Differences in Sequence Composition

The analysis involved examining each position along the sequences to determine the presence of mutations. In this study, the polymorphisms were defined as mutations present in ≥0.5% of group M sequences. By comparing each base to a reference sequence, we calculated the frequency of amino acid changes at specific positions as follows:$$f\left(i\right)=\frac{\sum_{j=1}^{N}I\;\left(x_{j}\neq a_{i}\right)}{L}$$
where *f*(*i*) represents the mutation frequency at the position *i*, *N* is the total number of sequences examined, *x_j_* is the amino acid at the position *i* in the sequence *j*, *I*(*x_j_* ≠ *a_i_*) is a function that returns 1 if *x_j_* ≠ *a_i_* and 0 otherwise, and *L* is the length of the reference sequence. This allowed us to identify positions of increased mutational activity, providing insight into the molecular evolution of the sequence. In addition to isolated positions, we constructed cumulative mutation profiles, illustrating the cumulative frequency of mutations over the entire length of the sequences. This facilitated the identification of regions characterized by higher mutation density, offering cues for potential functional domains or regions under selective pressure. 

Next, divergence analysis aimed to quantify dissimilarity between the reference sequence and those within specific subtypes, excluding ambiguous or non-contributing elements, to provide a comprehensive perspective on the genetic variability present. For each subtype, the divergence was calculated as: $$D=\frac{1}{N\times M}\overset{N}{\underset{i=1}{\sum }}\overset{M}{\underset{j=1}{\sum }}d_{ij}\times 100$$
where *D* represents the average divergence, *N* and *M* are the total number of sequences, and *d_ij_* represents the divergence between the sequences *i* and *j*.

By comparing each sequence to a reference, we calculated the number of amino acid variations, describing the specific nature of genetic divergence at the protein level. The results were saved in a divergence matrix, which systematically captures the dissimilarity between each sequence pair. This matrix facilitated the identification of clusters of sequences with similar divergence profiles, contributing to a more granular understanding of genetic relationships. Finally, to test the statistical significance of the observed mutation frequencies, hypothesis tests, such as the chi-square test, were applied to determine whether the observed mutation patterns deviated significantly from what would be expected by chance alone. This step added a level of robustness to our results, ensuring that the identified mutation frequencies were not simply stochastic fluctuations. All statistical analyses were performed using R (version 4.3.1). A *p*-value of <0.05 was considered statistically significant.

## 3. Results

### 3.1. Distribution of Published HIV Integrase Sequences in SSA

Figure 1 shows the distribution of published sequences of HIV IN across SSA collected over forty years and included in the study.

We examined a total of 7906 HIV IN gene samples, representing 3% (7906/238, 924) of the HIV-1 sequences available in public databases. These samples were specifically selected for analysis because they corresponded to the gene of interest, with the remaining sequences being excluded. Our analysis encompassed integrase sequences across various HIV-1 subtypes from 1983 to 2023, including subtype A (24.4%, 1932/7906), B (0.17%, 14/7906), C (57.23%, 4525/7906), D (4.32%, 342/7906), F (0.59%, 47/7906), G (1.2%, 95/7906), H, J, K, L (0.21%, 17/7906), CRFs (10.62%, 840/7906), and the untyped groups N, O, P, and U (1.21%, 96/7906). While Figure 1 does not report the distribution of HIV subtypes and recombinant forms within the sub-region of SSA over time, it does depict the integrase samples included in the study. The sampling of HIV subtypes in Figure 1 is in agreement with previous reports concerning the HIV genetic diversity in SSA [1,2,3,4].

According to HIV-1 molecular epidemiology studies in the African continent [1,2,3,4], the majority of IN samples from 22 countries in SSA since the onset of the epidemic were of subtypes A, B, C, D, G, H, and L. From 1993 onwards, sequences of subtypes F, J, K, CRFs, and the untyped groups N, O, and U were also documented. Subtype C samples have been consistently significant throughout the entire period, especially in Southern and Eastern Africa and, to a lesser extent, in Central Africa. In Eastern Africa, subtypes A and D samples were predominant from 1990 to 2020. In contrast, Western Africa had limited representation, mainly with a few sequences from subtype G and CRFs. Central Africa showcased a diverse range of sequences from all subtypes and HIV groups throughout the period, with a notable expansion of CRFs between 2019 and 2022.

### 3.2. HIV Integrase Amino Acid Diversity

Table 1 shows the extent of IN amino acid diversity by groups and subtypes for sequences from two time periods, 1983–2007 and 2008–2023, using subtype B (HXB2) as the reference strain. 

Statistical analysis did not show significant differences in IN amino acid diversity between the years 1983 and 2007 and 2008 and 2023. However, an increase in amino acid diversity was found in subtypes A (0.06 ± 0.03 vs. 0.07 ± 0.04) and D (0.04 ± 0.02 vs. 0.06 ± 0.05), and notably in the NOPU group (0.12 ± 0.07 vs. 0.17 ± 0.19), while a decrease was noted in subtypes B (0.05 ± 0.03 vs. 0.04 ± 0.03) and C (0.06 ± 0.02 vs. 0.05 ± 0.02) between the years 1983 and 2007 and 2008 and 2023. For subtypes H, J, K, and L, amino acid diversity (0.06 ± 0.03) was estimated only for the 1983–2007 period. No variation was found in subtypes F, G, CRFs, and all HIV groups (M, N, O, P) and U between the two sampling periods. Overall, these results were in agreement with previous estimates of mean amino acid diversity, around 7% between HIV sub-subtypes and 0.11–0.18% between HIV groups [9]. Regarding subtypes A and D, the observed divergences between the years 1983 and 2007 and 2008 and 2023 suggested that there may have been a trend toward a genetic diversification over time, which was also much more evident in the NOPU group, that could shape INSTI effectiveness and INSTI DRM in SSA. 

### 3.3. Natural Polymorphism Patterns and Conservation Analysis

To evaluate the conservation degree in INSTI-DRMs and polymorphisms of HIV-IN amino acidic sequences in SSA over time, we conducted a chronological analysis of the overall contribution of the entire study population. The dataset was divided into two time periods, 1983–2007 (3491 samples) vs. 2008–2023 (4415 samples), with consideration given to the time before and after approval of INSTI-based treatment regimens [13], and the amino acidic sequences were first aligned and compared to the reference strain B (HXB2) and then compared to each other (Figure 2).

The polymorphisms, defined as mutations present in ≥5% of sequences, were found in 49/288 (17.0%) and 54/288 (18.75%) amino acid positions in the sequences from the 1983–2007 and 2008–2023 periods, respectively. Highly polymorphic positions, defined as positions with substitutions detected in ≥20% of sequences analyzed, were lower in samples from 1983–2007, accounting for 7.29% (21/288) of positions, compared to those from 2008–2023, where they accounted for 9.02% (26/288). Some positions (K14, D25, V31, M50, F100, L101, T112, T124, T125, G134, K136, D167, V201, T206, T218, L234, A265, R269, D278, and S283) remained highly polymorphic across both periods, while other positions (E11, S17, I113, S119, and D256) were highly polymorphic only from 2008–2023.

Concerning highly conserved amino acid positions (with <1% variability), sequences between 1983 and 2007 and 2008 and 2023 showed similar levels of variability, accounting for 65.62% (189/288) and 64.63% (187/288), respectively. Among the 288 IN amino acids, there was no significant difference in variability concerning major or accessory INSTI DRMs, except for position G163, which was less conserved in samples from 2008–2023.

### 3.4. Prevalence of INSTI Major DRMs

The overall estimated prevalence for major INSTI-DRMs over forty years was 0.35% (28/7906), comprising 0.51% (18/3491) in the period of 1983–2007 and 0.22% (10/4415) in the period of 2008–2023. Notably, there was a statistically significant decrease (*p* = 0.03) observed in 2008–2023 as compared to 1983–2007. Among the major INSTI-DRMs identified, the most common was T66A, accounting for 0.08% (6/7906), followed by Y143H and E138K at 0.05% (4/7906) each, G118R and S147G with 0.04% (3/7906) each, and N155D at 0.03% (2/7906). Less frequently E92G, E138A, G140R, Q148H, N155H, and N155T were observed at 0.01% (1/7906) each. No major DRMs were found in subtypes B, D (1983–2007), F, G, HJKL, CFR_S_ (2008–2023), or the NOPU group. Further details regarding the prevalence of major DRMs over the two time periods, as per the Stanford HIVDB 8.9-1, can be found in Table 2.

### 3.5. Prevalence of INSTI Accessory DRMs

Overall, the prevalence of INSTI accessory resistance mutations over the course of forty years was 46.61% (2665/7906), with a rate of 22.26% (1760/7906) in 1983–2007 samples vs. 24.29% (1921/7906) in 2008–2023 samples (as shown in Table 2). Among the identified INSTI accessory mutations, the most common were M50I at 33.70% (2665/7906), followed by L74I at 6.89% (544/7906), T97A at 2.77% (219/7906), E157Q at 1.59% (126/7906), and L74M at 1.23% (98/7906). Less-prevalent mutations included Q95K at 0.20% (16/7906), G163R at 0.1% (8/7906), D232N at 0.088% (7/7906), and G163K at 0.02% (2/7906).

The prevalence of residue substitutions in the 1983–2007 and 2008–2023 periods was as follows: M50I 35.40% (1236/3491) vs. 32.36% (1429/4415), L74I 6.87% (240/3491) vs. 6.88% (304/4415), Q95 0.22% (8/3491) vs. 0.18% (8/4415), T97A 4.2% (147/3491) vs. 1.63% (72/4415), E157Q 1.91% (67/3491) vs. 1.33% (59/4415), L74M 1.48% (52/3491) vs. 1.0% (46/4415), D232N 0.2% (7/3491), G163R 0.09% (3/3491) vs. 0.02% (1/4415), and G163K 0.04% (2/4415). In the NOPU group, M50I and L74I were statistically significantly more prevalent in the years 2008–2023 than in 1983–2007 (*p* < 0.05), while in subtype A, mutation T97A was statistically significantly more prevalent in the years 1983–2007 (*p* < 0.05). Mutation D232N was detected only in samples from the years 1983–2007 (see Table 3).

## 4. Discussion

Limited information is available regarding the evolution of INSTI-DRM resistance in SSA, underscoring the presence of pre-existing DRMs before transitioning to dolutegravir-based ART regimens, which could significantly impact virological outcomes [14,15]. Several groups have reported contrasting data on INSTI-DRMs in SSA, with many showing a lack of major DRMs and a low frequency of accessory INSTI-DRMs [15,16,17], while others reported frequencies ranging from 0.8% up to 5.4% of major INSTI-DRMs in INSTI-naïve populations, suggesting a potential threat to the effectiveness of the DTG-based ART programs in SSA [18,19,20].

Our large-scale analysis provided a comprehensive mapping of polymorphisms and conserved amino acid positions within the HIV IN gene across major HIV-1 subtypes and HIV groups, considering both INSTI-naïve and experienced individuals. This analysis also examined the baseline INSTI resistance, encompassing major and accessory DRMs, with implications for the use of DTG as a first-line regimen.

HIV-1 genetic diversity in SSA plays a crucial role in susceptibility to INSTIs and the selection of resistance mutations [18,19,20,21,22]. In our study, the majority of the viruses analyzed were subtypes C and A, followed by CRFs, D, F, and G, as well as groups N, O, P, and U, whereas subtypes H, J, F, L, and B were poorly represented, reflecting the distribution of HIV infection in SSA. Despite a significant increase in the number of CRFs and changes in the geographic distribution of M HIV-1 subtypes and groups N, O, P, and U forms over the past forty years, the diversity estimates for IN inter-subtype diversity levels remained consistently low at <0.07, as did inter-group diversity at <0.18, during both the 1983–2007 and 2008–2023 periods. This suggested that the IN region is a conserved drug target [9].

Analysis of the conservation within the genetically conserved key function enzyme amino acids revealed similar levels of variability (<1%) in both the 1983–2007 and 2008–2023 periods, consistent with findings in HIV-1 group M subtypes and previous studies [9,18]. In contrast, a higher number of polymorphic positions were detected at a frequency of ≥5% or ≥20% in samples from the 2008–2023 period. Notably, residue G163, where substitutions have been reported to act synergistically with other mutations to reduce susceptibility to INSTIs, appeared to be more polymorphic (≥5% variability) in samples from 1983–2007 [23].

Estimates of INSTI-DRMs are limited within SSA [24,25,26,27,28,29,30,31,32,33,34,35,36,37,38]. Most reports indicate that primary INSTI DRMs are rare, and very few INSTI accessory resistances can be detected prior to the initiation of INSTI therapy, possibly due to transmitted INSTI resistances and the cumulative natural occurrence of mutations without selective drug pressure [27,39]. DRMs such as G118R, E138K/A/T, G140S/A/C/R, Q148H/R/K, S153 F/Y, N155H, and R263K have been identified as being associated with DTG resistance [34]. In our study, the examination of INSTI resistance-associated mutations showed low levels of major DRMs, such as T66A, E92G, G118R, E138K/A, G140R, Y143H, S147G, Q148H, and N155H/T/D, over forty years, and as mentioned above, some of these have been reported to have an effect on DTG. Notably, it was unexpected to detect major INSTI-DRMs at statistically significantly higher frequencies in 1983–2007 compared to 2008–2023. The most common major DRM was T66A, found in Cameroon, Rwanda, and South Africa [38]. The rare mutation E92G was present in South Africa; G118, previously described as a DTG-resistance pathway in non-B subtypes, was present in South Africa; the extremely rare mutation G140R was present in South Africa; and the less common mutation Y143H was present in Uganda and Botswana. The appearance of INSTI resistance mutations in the pre-INSTI era, as found in our study, might be explained by low retention in care and poor treatment adherence in SSA, while the decline in the frequency of DRMs in the INSTI era likely reflects the availability of ART regimens and improved adherence in treatment-experienced populations [37,39].

Regarding accessory DRMs, in spite of the broad genetic diversity over forty years, no relevant difference in mutation frequency between 1983–2007 and 2008–2023 was detected, and more than 46% of the study population harbored at least one mutation. Seven accessory INSTI-DRMs that may contribute to drug resistance when combined with major DRMs were found, with M50I, L74I/M, T97A, and E157Q being the most frequent. In particular, M50I, L74I/M, and T97A, in combination with other INSTI drug resistance mutations, may contribute to reduced DTG susceptibility [37,40]. These mutations, which have been shown to occur before initiation of cART with different prevalences depending on the subtypes, could facilitate viral evolution of resistance under drug-selective pressure. M50I has been shown to increase DTG resistance in combination with R263K [39]. Here, we show a high prevalence of the M50I in HIV subtypes circulating in SSA over forty years, particularly in subtype C and in HIV groups N, O, P, and U, with a higher increase in groups N, O, P, and U in 2008–2023. L74 is located in the catalytic core domain (amino acids 51–212) and is involved in the catalytic processes of the enzyme. L74M, in combination with T79A, may reduce susceptibility to RAL and/or EVG in the absence of primary INI-resistance mutations [40]. It has also been reported that L74M combined with G118R or E138K mutations increases resistance to DTG [41]. To date, the role of L74I substitution is still unclear [42,43], although it has been shown to enhance integrase inhibitor resistance in combination with additional major INSTI mutations [8] and is also being evaluated as a putative cause of DRMs for ART using DTG [41]. Our analysis indicated a low prevalence of L74I polymorphisms in subtypes A, D, and F, a relatively high prevalence in subtypes C, G, and CRFs, and a high prevalence in groups N, O, P, and U, particularly in 2008–2023, while L74M remained relatively low except in CFRs, where it reached a prevalence of about 6%.

T97A is a relatively common polymorphism-associated INSTI resistance and is co-selected in the presence of major INSTI-DRMs by RAL and EVG and by DTG in INSTI-experienced individuals [44]. Our analysis revealed that in subtype A, mutation T97A had a statistically significantly higher prevalence in the years 1983–2007 compared with 2008–2023, and it had a relatively high prevalence in subtype F and the NOPU group, while it was absent in the NOPU group in the years 2008–2023. Our results suggest the existence of major and accessory DRMs prior to INSTI-based therapy that poses a minimal risk of compromised virologic response for patients enrolling in INSTI-based therapy.

This study is limited in that (a) our analysis did not distinguish ART-experienced from treatment-naïve individuals in the years 1983–2007 and 2008–2023, (b) the heterogeneity in the number of sequences analyzed for different African regions depends on the number of sequences deposited in the Los Alamos HIV database, (c) the limited number of available IN sequences for subtypes F and G and NOPU groups may have affected the frequency of DRMs and polymorphisms at specific positions.

## 5. Conclusions

In summary, our study presented a comprehensive analysis of HIV integrase gene variability over forty years. Our data showed a higher prevalence of major DRMs in the years 1983–2007, prior to INSTI-based therapy, than in the years 2008–2023. This could be explained by the low retention in care and the poor treatment adherence in SSA, which favor genetic variability and the high evolution rate of HIV-1. No significant difference in mutation frequency was detected in accessory DRMs in the years 1983–2007 and 2008–2023. Natural HIV-1 variation could explain this because accessory DRMs occur in the absence of therapy, but similarly to major DRMs, other factors, such as poor adherence support and poor retention in care, may also support the level of resistance reported here. The results of this study suggest the need to expand and strengthen surveillance through an HIV drug resistance national survey.

## Figures and Tables

**Figure 1 pathogens-13-00102-f001:**
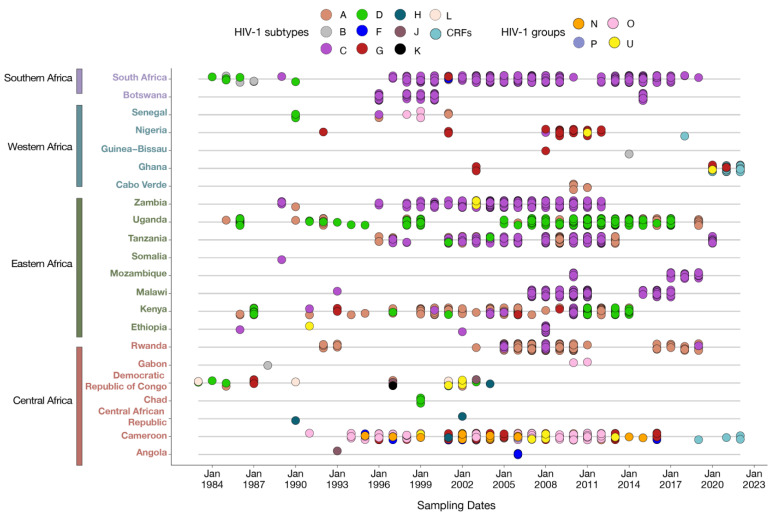
Temporal sampling of HIV integrase sequences in different African regions (1983–2022). Sampling of all available HIV IN sequences in different African countries through time, annotated according to their subtype assignment. Each dot represents an individual sample, with overlapping subtypes indicating isolation within the same time frame.

**Figure 2 pathogens-13-00102-f002:**
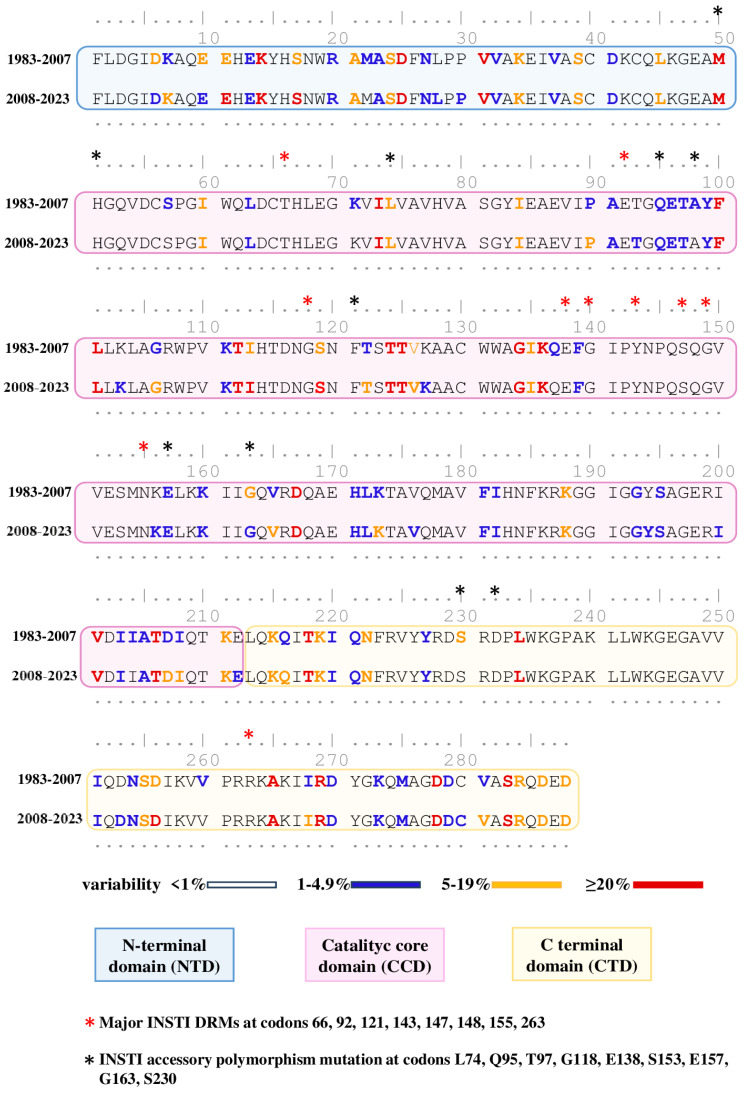
Comparison of degree variability between HIV integrase sequences from 1983–2007 vs. 2008–2023. The reference strain B (HXB2), used for the definition of mutations, is colored according to the frequency rate of mutations present in the 1983–2007 and 2008–2023 groups. Domains of IN are colored boxes. Red asterisk indicates major INSTI DRMs, and black asterisk indicates INSTI accessory polymorphic and nonpolymorphic mutations from the latest version of Stanford’s list (https://hivdb.stanford.edu/, accessed on 25 November 2023).

**Table 1 pathogens-13-00102-t001:** The inter-group–subtype diversity of IN sequences (%).

Divergence	(1983–2007)		(2008–2023)
Subtype A (618) vs. B	0.06 ± 0.03	Subtype A (1314) vs. B	0.07 ± 0.04
Subtype B (11) vs. B	0.05 ± 0.03	Subtype B (3) vs. B	0.04 ± 0.03
Subtype C (2099) vs. B	0.06 ± 0.02	Subtype C (2024) vs. B	0.05 ± 0.02
Subtype D (148) vs. B	0.04 ± 0.02	Subtype D (194) vs. B	0.06 ± 0.05
Subtype F (32) vs. B	0.05 ± 0.02	Subtype F (15) vs. B	0.05 ± 0.03
Subtype G (51) vs. B	0.05 ± 0.01	Subtype G (44) vs. B	0.05 ± 0.02
Subtypes (H, J, K, L) (17) vs. B	0.06 ± 0.03	Subtypes (H, J, K, L) vs. B	-
CRF_s_ (453) vs. B	0.06 ± 0.02	CRF_s_ (387) vs. B	0.06 ± 0.02
Groups (N, O, P, U) (62) vs. B	0.12 ± 0.07	Groups (N, O, P, U) (34) vs. B	0.17 ± 0.19
All groups (M, N, O, P, U) (3491) vs. B	0.07 ± 0.04	All groups (M, N, O, P, U) (4415) vs. B	0.07 ± 0.04

Divergence: the mean proportion of amino acid difference between all sequence pairs. The number of sequences compared are in brackets.

**Table 2 pathogens-13-00102-t002:** Distribution of major DRMs across subtypes.

	A 1983–2007 [618]	A 2008–2023 [1314]	C 1983–2007 [2099]	C 2008–2023 [2426]	D 1983–2007 [148]	CFR_S_ 1983–2007 [453]
T66A	0.32 A(2)		0.1 A(2)	0.04 A(1)		0.22 A(1)
E92G			0.08 G(1)			
G118R				0.12 R(3)		
E138K/A		0.15K(2)	0.05 A(1)	0.04 K(1)	0.6 K(1)	
G140R				0.12 R(1)		
Y143H			0.1 H(2)	0.04 H(1)	0.68 H(1)	
S147G			0.1 G(2)	0.04 G(1)		
Q148H			0.05 H(1)			
N155H/T/D			0.05 D(2)		0.68 H(1)	0.22 T(1)

Frequency of major DRMs, n (%), is shown. Round brackets: number of samples harboring the variant. Square brackets: number of samples tested for each subtype.

**Table 3 pathogens-13-00102-t003:** Distribution of INSTI accessory resistance across subtypes.

	A	A	C	C	D	D	F	F	G	G	HJIL	CFR_S_	CFR_S_	NOPU	NOPU
	1983-	2008-	1983-	2008-	1983-	2008-	1983-	2008-	1983-	2008-	1983-	1983-	2008-	1983-	2008-
	2007	2023	2007	2023	2007	2023	2007	2023	2007	2023	2007	2007	2023	2007	2023
	[618]	[1314]	[2099]	[2426]	[148]	[194]	[32]	[15]	[51]	[44]	[17]	[453]	[387]	[62]	[34]
M50I	18.28	9.36	47.97	48.8	0.67	2.58	18.75	26.6	11.74	18.18		17.66	19.9	*** 37.09**	*** 85.29**
	I(113)	I(123)	I(1007)	I(1184)	I(1)	I(5)	I(6)	I(4)	I(6)	I(8)		I(80)	I(77)	**I(23)**	**I(28)**
L74M/I	5.82	3.50	6.29	6.00	4.00	2.58	3.10	6.60	15.69	18.18		15.45	17.57	*** 59.67**	*** 88.24**
	I(36)	I(46)	I(83)	I(146)	I(6)	I(5)	M(1)	I(1)	I(8)	I(8)		I(70)	I(68)	**I(37)**	**I(30)**
	0.97	1.67	0.76	0.08	0.67				1.96	2.27		6.00	5.42		
	M(6)	M(22)	M(16)	M(2)	M(1)				M(1)	M(1)		M(27)	M(21)		
Q95K	0.32	0.15	0.14	0.16		0.51				2.27	5.88	0.44			
	K(2)	K(2)	K(3)	K(4)		K(1)				K(1)	K(1)	K(2)			
T97A	*** 15.04**	*** 3.27**	0.86	0.28	2.68		12.50	6.60	5.88	6.82	5.88	3.97	4.65	9.60	
	**A(93)**	**A(43)**	A(18)	A(7)	A(4)		A(4)	A(1)	A(3)	A(3)	A(1)	A(18)	A(18)	A(6)	
E157Q	1.13	2.28	2.00	0.28	2.68					2.27		3.00	5.42		
	Q(7)	Q(30)	Q(42)	Q(7)	Q(4)					Q(1)		Q(14)	Q(21)		
G163R/K	0.16	0.07	0.05	0.16	0.67								0.25		
	R(1)	K(1)	R(1)	R(4)	R(1)								K(1)		
													0.25		
													R(1)		
D232N			0.19		1.35							0.22			
			N(4)		N(2)							N(1)			

Frequency of accessory DRMs, n (%), is shown. Round brackets: number of samples harboring the variant. Square brackets: number of samples tested for each subtype; ***** NOPU group M50I and L74I (1983–2007) vs. (2008–2023), subtype A, T97 (1983–2007) vs. (2008–2023) (*p* < 0.05).

## Data Availability

The sequences analyzed for this study are available at Los Alamos National Laboratory “https://www.hiv.lanl.gov/content/index”.

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
