# Peer review of "Comprehensive Analysis of HIV-1 Integrase Resistance-Related Mutations in African Countries"

_pathogens, 2024, doi:10.3390/pathogens13020102_

Round 1

Reviewer 1 Report

Comments and Suggestions for Authors

The authors conducted a comprehensive analyses of integrase gene sequences in Africa over 40 years, pre and prior INSTI therapy. These analyses are extremely crucial due to the increased resistance in PLWH in Africa on NRTI or NNRTI or PI regimens and the switching to INSTIs, specifically, DTG, after the roll-out of FDA-approved INSTIs. However, drug resistance surveillance and monitoring of INSTI mutations are most likely nonexistent and pressing in most settings to prevent treatment failures and drug resistant mutant emergence as seen previously with other classes of antiretrovirals. The authors do a great job of presenting and introducing the problems in the introduction, describing the materials and methods, and summarizing the results and describing the implications of the study in the discussion. However, the beginning of the results is extremely difficult to follow, and the figures/body of the manuscript need significant rewriting and revision. However, tables 2 and 3 are the most striking data and best to understand and follow. I could see the omission of section 3.2, I’m not sure how relevant it is to the resistance story. Figure 2 is difficult to follow and must be broken down by subtype so not sure how this could be simplified in its current format.

Rewrite 34-35 “with Africa, with…”

46- please add non-nucleoside reverse transcriptase inhibitors

85- delete hyphen in the word regions

142- shouldn’t this be figure 1, not table 1. Please explain figure 1 more, why are some subtypes overlaid together? This needs to explain in great detail, nothing is noted about the years and percentages in the results..or how one subtype dominated in a given year versus another in other years in a particular region..

162- table 1 data, I can see some differences, however, could you elaborate further on how these differences are significant for subtypes A-D, I can see this difference for NOPU group?

196- is figure 2 connected to section 3.3, please indicated in body of manuscript

369- please review L74I statement, there have been several examples in vitro where reductions of potency for several INSTIs has been observed again IN mutants with this mutation in combination with other mutations.

Author Response

We thank the Reviewers for their criticisms and suggestions. The manuscript has been revised accordingly and additional points are discussed below. For the Reviewers’ convenience, the revisions have been highlighted in the manuscript.

The authors conducted a comprehensive analyses of integrase gene sequences in Africa over 40 years, pre and prior INSTI therapy. These analyses are extremely crucial due to the increased resistance in PLWH in Africa on NRTI or NNRTI or PI regimens and the switching to INSTIs, specifically, DTG, after the roll-out of FDA-approved INSTIs. However, drug resistance surveillance and monitoring of INSTI mutations are most likely nonexistent and pressing in most settings to prevent treatment failures and drug resistant mutant emergence as seen previously with other classes of antiretrovirals. The authors do a great job of presenting and introducing the problems in the introduction, describing the materials and methods, and summarizing the results and describing the implications of the study in the discussion. However, the beginning of the results is extremely difficult to follow, and the figures/body of the manuscript need significant rewriting and revision. However, tables 2 and 3 are the most striking data and best to understand and follow. I could see the omission of section 3.2, I’m not sure how relevant it is to the resistance story. Figure 2 is difficult to follow and must be broken down by subtype so not sure how this could be simplified in its current format.

Re: We apologize for the lack of clarity, and we thank the Reviewer for the suggestion.

Figure 1 does not report the distribution of HIV subtype sequences available in public database across SSA over time but rather only integrase sample distribution included in the study. For this reason, it was not possible to provide prevalence data of subtypes in different regions of Africa, based on the availability in public repository of genome sequences from this genomic region. Despite this, sampling of integrase sequences in Africa over 40 years shown in Figure1 is in agreement with the distribution of subtypes previously published by us and other authors.  As requested by the Reviewer, a better description of the results shown in Figure 1 has been included in the paragraph.

Regarding section 3.2, taking respectfully into consideration the Reviewer’s observation, we think that analysis of HIV subtype divergence can delineate a genetic diversification over time that could shape INSTI effectiveness and INSTI DRM.

Concerning Figure 2, we agree in part with the Reviewer’s criticism. The degree of variability/conservation of integrase sequences from 1983-2007 vs 2008-2023 was overall analysed by comparing first the dataset of 3491 samples and the dataset of 4415 samples with the reference strain B (HXB2) e then and then comparing the datasets with each other. This analysis does not distinguish the different subtypes but is carried out on the total samples in the two different time periods, which were then compared with each other. For better understanding of the text, Figure 2 has been modified and simplified and we hope it now meets with the Reviewer’s demand. 

Rewrite 34-35 “with Africa, with…”

Re: The text have been revised according to the Reviewer suggestions.

46- please add non-nucleoside reverse transcriptase inhibitors

Re: Thank you for this observation. The text have been revised accordingly.

85- delete hyphen in the word regions

Re: This has been changed accordingly.

142- shouldn’t this be figure 1, not table 1. Please explain figure 1 more, why are some subtypes overlaid together? This needs to explain in great detail, nothing is noted about the years and percentages in the results or how one subtype dominated in a given year versus another in other years in a particular region.

Re: We thank the reviewer for noticing the error. Table 1 has been changed in Figure 1 accordingly.

Regarding Figure 1, we apologize with the Reviewer for the lack of clarity. For determination of HIV-1 integrase sampling distribution in SSA were analyzed a total number of 7906 samples, that represent only 3% of all gene sequences available in Los Alamos database. Despite that, the distribution of subtypes is in agreement with our previous work concerning the HIV genetic diversity in SSA (Giovanetti M.et al.2020).

While Figure 1 does not report the distribution of HIV subtypes and recombinant forms within the sub-region of SSA over time it does depict the integrase samples included in the study. Each dot represents an individual sample, with overlapping subtypes indicating isolation within the same time frame. As requested by the Reviewer, the text has been implemented accordingly in the new version of the manuscript. We hope that the new text meets the Reviewer’s demands.

162- table 1 data, I can see some differences, however, could you elaborate further on how these differences are significant for subtypes A-D, I can see this difference for NOPU group?

Re: We thank the reviewer for raising this issue. Although subtypes A-D and NOPU group displayed an increase in amino acid diversity, no statistically significant differences in integrase amino acid diversity between the years 1983-2007 and 2008–2023 were found. To respond to the reviewer’s concern, we expanded and improved the section regarding this important point accordingly.

196- is figure 2 connected to section 3.3, please indicated in body of manuscript.

Re: We thank the reviewer for noticing the error.

369- please review L74I statement, there have been several examples in vitro where reductions of potency for several INSTIs has been observed again IN mutants with this mutation in combination with other mutations.

Re: We thank the Reviewer for bringing up this point. The paragraph regarding L74I mutation has been rewritten and detailed information regarding reductions of potency for INSTIs has been included.

Reviewer 2 Report

Comments and Suggestions for Authors

Comprehensive analysis of HIV-1 integrase resistance-related mutations in African countries. Branda, et al.

The authors present a timely and important analysis of drug resistance mutations for HIV-1 integrase in African, divided into four regions. The analysis included 40 years of sequence analysis of integrase that were divided into periods prior to the introduction of INSTs (1983-2007) and after (2008-2023).

Comments:

Figure 1 is especially important to identify specific groups with different regions of Africa. It is difficult to visualize because of the light colors used in naming the countries, specifically the exceptionally light greenish colors. Possibly bold these countries names. If the graph size could be increased, the different colored circles could be enlarged making it more informative. Better visualization would be great for the reader to follow all the data in the text. The rest of the figures and Tables appear proper.

The presentation of data, English writing and the discussion appear very appropriate. 

 Minor:

There are typos on line 85 and 358.

Author Response

We thank the Reviewers for their criticisms and suggestions. The manuscript has been revised accordingly and additional points are discussed below. For the Reviewers’ convenience, the revisions have been highlighted in the manuscript.

Figure 1 is especially important to identify specific groups with different regions of Africa. It is difficult to visualize because of the light colors used in naming the countries, specifically the exceptionally light greenish colors. Possibly bold these countries names. If the graph size could be increased, the different colored circles could be enlarged making it more informative. Better visualization would be great for the reader to follow all the data in the text. The rest of the figures and Tables appear proper.

Re: We appreciate the reviewer's comment, and we have made the necessary adjustments to the color scale as per your suggestion. However, with regards to increasing the circle size, we must express our constraint in doing so. Summarizing a large volume of data presents a significant challenge, and further increasing the circle size could lead to an overlap of tips, making it difficult for readers to comprehend the analysis.

The presentation of data, English writing and the discussion appear very appropriate.

Re: We thank the Reviewer for his/her positive assessment of our manuscript

There are typos on line 85 and 358.

Re: We thank the reviewer for noticing the error.

Round 2

Reviewer 1 Report

Comments and Suggestions for Authors

The authors' revisions are acceptable, and the manuscript is greatly improved and merits publication.